# Anti-Obesity Medications and the Risk of Obesity-Related Cancers in Older Women: A Propensity Score Matching Analysis of 2007–2015 SEER-Medicare Data

**DOI:** 10.3390/cancers17101624

**Published:** 2025-05-11

**Authors:** Omer Abdelgadir, Maryam R. Hussain, Kelseanna Hollis-Hansen, Carlos H. Barcenas, Yong-Fang Kuo, Celette S. Skinner, Lindsay G. Cowell, Sarah E. Messiah, David S. Lopez

**Affiliations:** 1Graduate School of Biomedical Science, University of Texas Medical Branch, Galveston, TX 77555, USA; 2O’Donnell School of Public Health, University of Texas Southwestern Medical Center, Dallas, TX 75390, USA; kelseanna.hollis-hansen@utsouthwestern.edu (K.H.-H.); celette.skinner@utsouthwestern.edu (C.S.S.); lindsay.cowell@utsouthwestern.edu (L.G.C.); sarah.messiah@utsouthwestern.edu (S.E.M.); david.lopez3@utsouthwestern.edu (D.S.L.); 3School of Public and Population Health, University of Texas Medical Branch, Galveston, TX 77555, USA or marhussain@salud.unm.edu (M.R.H.); yokuo@utmb.edu (Y.-F.K.); 4Department of Internal Medicine, University of New Mexico Health Sciences Center, Albuquerque, NM 87131, USA; 5Simmons Comprehensive Cancer Center, University of Texas Southwestern Medical Center, Dallas, TX 75235, USA; 6Department of Breast Medical Oncology, University of Texas MD Anderson Cancer Center, Houston, TX 77230, USA; chbarcenas@mdanderson.org

**Keywords:** anti-obesity medication, phentermine, liraglutide, obesity-related cancer, breast cancer, colorectal cancer, ovarian cancer, endometrial cancer

## Abstract

Obesity rates in the U.S. have reached alarming levels (42% of adults). Severe obesity (BMI ≥ 40 kg/m^2^) affects 9.2% of the population, more prevalent in women. This increases the risk of obesity-related cancers (ORCs), including breast (BrCa), colorectal (CRC), endometrial (ECa), and ovarian (OCa) cancers in women. Anti-obesity medications (AOMs) are becoming more commonly used to help with weight loss. This study explored whether using AOM is associated with a reduce risk of ORCs in older women. We found that AOM use, including phentermine, was associated with a reduced risk of ORCs, advanced-stage ORCs, BrCa, CRC, and advanced-stage CRC in older women. Notably, phentermine use was specifically associated with a reduced risk of BrCa and ECa. These findings could support clinical trials to evaluate phentermine as a potential preventive strategy for BrCa and ECa in high-risk older women, including those with obesity, genetic factors, and hormone replacement therapy use.

## 1. Introduction

Obesity is a chronic disease characterized by significant hormonal and metabolic disruptions. Beyond excessive body weight, obesity is linked to alterations in sex hormone metabolism, insulin and insulin-like growth factor (IGF) signaling, adipokines, and inflammatory pathways [1,2]. Key health outcomes associated with obesity include hyperglycemia, hyperinsulinemia, dyslipidemia, chronic low-grade inflammation, elevated levels of adipokines, leukocytosis, neutrophilia, and increased neutrophil extracellular traps [3,4,5,6]. These biomarkers have been implicated in the initiation and progression of various obesity-related cancers (ORCs), including breast cancer (BrCa), colorectal cancer (CRC), endometrial cancer (ECa), and ovarian cancer (OCa) [1,2,3,4,5,6,7].

The prevalence of obesity has reached alarming levels in the United States, affecting 42% of adults. Moreover, the age-adjusted prevalence of severe obesity (a body mass index [BMI] of 40 kg/m^2^ or higher) is 9.2% and is higher in women than in men [8]. According to the American Cancer Society’s 2025 report, an estimated 316,950 new cases of BrCa, 71,810 of CRC, 69,120 of ECa, and 20,890 of OCa were expected in women [9]. While not all cases are attributable to obesity, these cancers have been associated with excess body weight in the literature [1,2,3,4,5,6,7].

Weight management strategies, such as lifestyle modifications, metabolic and bariatric surgery, and the use of anti-obesity medications (AOM), are crucial for tackling the obesity epidemic [10]. Earlier AOMs, such as phentermine, liraglutide, diethylpropion, lorcaserin, orlistat, and phendimetrazine, have gained popularity in obesity management, mainly due to their low cost [11]. These medications act through distinct mechanisms. Phentermine, diethylpropion, and phendimetrazine function as sympathomimetic appetite suppressants. Orlistat works by inhibiting fat absorption in the gastrointestinal tract. Liraglutide, a glucagon-like peptide-1 receptor agonist (GLP-1 RA), helps regulate appetite and food intake, while lorcaserin targets serotonin 2C receptors to promote satiety [12]. Their effectiveness in promoting weight loss is well documented. While our study focuses on earlier AOMs, it is important to acknowledge the growing use of newer GLP-1 RA therapies, such as semaglutide and tirzepatide. These agents have demonstrated substantial efficacy in promoting weight loss and improving metabolic control, and they are reshaping current approaches to obesity management [13].

However, the long-term effects of the earlier AOMs on cancer outcomes remain inconclusive. In 2020, the U.S. Food and Drug Administration (FDA) highlighted an increased cancer risk associated with lorcaserin [14]. Whereas some research indicated a possible link between liraglutide and thyroid cancer [15,16], a meta-analysis did not validate this association [17]. Although research has consistently demonstrated that natural weight loss can mitigate the risk of ORCs, the specific impact of AOM on the risk of ORCs in older women remains understudied [18,19,20]. Given the escalating obesity rates, especially severe obesity among women, and the growing popularity of AOMs, exploring the association between AOM and ORCs in an older population of women is imperative. This study sought to address this knowledge gap by investigating the association between AOM use and the risk of ORCs in women aged ≥65 years.

## 2. Materials and Methods

### 2.1. Data Source

We conducted a retrospective analysis using the 2007–2015 Surveillance, Epidemiology, and End Results (SEER)-Medicare linked database, which integrates cancer registry data from 19 SEER regions with Medicare administrative claims [21]. We extracted data from the 5% non-cancer sample in the Summarized Denominator file. The SEER program collects clinical, demographic, and survival information on U.S. patients aged ≥65 years with cancer [22]. Medicare claims are linked through unique SEER identifiers covering the time of Medicare eligibility until death. This study was conducted in accordance with the Declaration of Helsinki and approved by the Institutional Review Board at the University of Texas Medical Branch (UTMB) at Galveston, TX, USA (Approval Code: 20-0237; Approval date: 15 September 2020). No patient informed consent specific to this study was required, as it involves secondary data analysis of de-identified, previously collected data.

### 2.2. Study Population

The study cohort included women aged ≥65 years (*n* = 576,565) with continuous Medicare Part D enrollment from July 2007 to June 2015. Patients were categorized as AOM-exposed or unexposed based on prescription data within this period. The index date for exposed individuals was the first AOM prescription date. We excluded those <65 years at index date, with <6 months of Medicare Parts A and B coverage prior to index date or diagnosed with ORCs (BrCa, CRC, ECa, OCa) within six months of index date. We matched each exposed patient to three unexposed patients (1:3) by birth year. For unexposed individuals, the index date was assigned based on the index date of their matched exposed counterpart to ensure comparable follow-up timeframes (matched cohort *n* = 160,230). All patients maintained ≥6 months of Medicare A and B coverage before the index date, ensuring comparability and mitigating immortal time bias. In addition, we mitigated potential selection bias by creating a 1:2 propensity score-matched (PSM) cohort for both the primary and secondary analyses based on AOM exposure (Figure 1).

### 2.3. AOM Exposure

The primary exposure in this study was the use of AOM, established from the Medicare Part D file using National Drug Codes (NDC) and Current Procedural Terminology (CPT) codes (available upon request). The AOMs included in this analysis were diethylpropion, liraglutide, lorcaserin, orlistat, phendimetrazine, and phentermine. To be classified as exposed, patients using AOM were required to have at least six months between the prescription date and the diagnosis of ORCs (BrCa, CRC, ECa, or OCa) (if any). Patients were categorized as AOM never-user (*n* = 7220, reference group) vs. AOM ever-user (*n* = 3610). For the secondary analysis, phentermine (never-user [*n* = 4430] vs. ever-user [*n* = 2214]) and liraglutide (never-user [*n* = 538] vs. ever-user [*n* = 269]) were evaluated separately as primary exposures, as they were the only AOM with sufficient sample sizes for individual analysis, using the same methodology as the primary analysis. On the other hand, patients who were classified as phentermine never-user or liraglutide never-user may have used other AOMs during the study period (Figure 1).

### 2.4. ORCs Outcomes

We examined the first-incident ORCs, advanced-stage ORCs, high-grade ORCs, BrCa, advanced-stage BrCa, high-grade BrCa, CRC, advanced-stage CRC, high-grade CRC, ECa, advanced-stage ECa, high-grade ECa, and OCa. These cancers were selected based on consistent associations with excess body fat identified by the International Agency for Research on Cancer (IARC) [23]. Incident ORCs was identified using International Classification of Diseases for Oncology-3 (ICD-O-3) site and histology codes from the SEER database. BrCa was defined by site codes C50.0–C50.9 and histology codes 8500–8543. CRC included site codes C18.0–C20.9 and histology codes 8140–8147, 8210–8211, 8220–8221, 8260–8263, 8480–8481, and 8490. ECa was defined by site codes C54.0–C55.9 and histology codes 8140, 8262, 8323, 8380–8384, 8480–8482, and 8560. OCa was identified using site code C56.9 and histology codes 8440–8469. Cancer staging was based on the American Joint Committee on Cancer (AJCC) system available in SEER. We assigned the first day of the cancer diagnosis month to enhance censoring accuracy, as SEER reports only the month and year of cancer diagnosis. Time-to-event was calculated in months starting from the index date (first AOM exposure). For patients who developed an ORCs, time-to-event was measured from the index date to the first ORCs diagnosis. Patients who did not develop ORCs were right-censored at the study end date (31 December 2015) or at the time of loss to follow-up, whichever occurred first. Patients who remained ORC-free and died before the study end date were right-censored at the time of death.

### 2.5. Covariates

Covariates included demographic and clinical characteristics, selected based on *a priori knowledge* [24]. Demographics included age at index date, race/ethnicity, and poverty status. Clinical variables included the NCI—Charlson Comorbidity Index (CCI) [25], diabetes, insulin and metformin use, hyperlipidemia, cardiovascular disease, malaise/fatigue, muscular wasting, hypogonadism, anterior pituitary dysfunction, depression, osteoporosis, Cushing’s syndrome, thyroid disorders, and polycystic ovary syndrome. We found no concerning multicollinearity among these predictors based on Variance Inflation Factor (VIF) analysis with cut-off value of VIF < 5 (Appendix A). All covariates were measured at least six months before the index date and the mandatory six-month gap between medication initiation and ORCs diagnosis date to mitigate ascertainment bias.

### 2.6. Propensity Score Matching (PMS)

We used PSM procedure to create comparable AOM groups with respect to observable characteristics and mitigate selection bias [26,27]. Using logistic regression, we first calculated propensity scores, representing the predicted probability of AOM use, for each subject based on a comprehensive set of covariates, including all patient demographics and clinical features previously described in the covariates section. We then conducted 1:2 greedy nearest neighbor matching without replacement algorithm to pair AOM never-users with similar propensity scores to AOM ever-users [27]. We assessed covariate balance using standardized mean differences, with values <0.10 indicating acceptable balance [28]. We replicated the PSM process for secondary analyses of phentermine and liraglutide.

### 2.7. Statistical Analysis

We summarized baseline characteristics using descriptive statistics. For categorical variables, we reported frequencies and proportions. For continuous variables, we calculated means and standard deviations (SDs). We compared study variables using *t*-tests, Chi-square tests, or Fisher’s exact tests, as appropriate. To evaluate ORCs outcomes in the 1:2 PSM cohorts, we applied conditional multivariable Cox proportional hazards models, accounting for clustering within matched pairs. We included age at the index date (first prescription) in all models to account for biological heterogeneity and reduce residual confounding. We assessed the proportionality assumption through log–log survival plots and weighted Schoenfeld residuals and evaluated model goodness-of-fit using Cox–Snell and deviance residuals. Additionally, we conducted stratified analyses for primary analysis (AOM) by diabetes status to explore potential effect modification. As a sensitivity analysis, we used Fine–Gray competing risk models to estimate subdistribution hazard ratios (sHRs) for both primary (AOM) and secondary analyses (phentermine and liraglutide), treating all-cause death as a competing event for incident ORCs. We considered *p*-values < 0.05 statistically significant. All statistical analyses were conducted using SAS (v.9.4, SAS Institute, Cary, NC, USA) and R (v.4.3.1, RStudio, Boston, MA, USA).

## 3. Results

Table 1 presents the baseline characteristics of women aged ≥65 years based on the use of AOMs before and after a 1:2 PSM. Mean age of the overall cohort was 76.1 ± 6.9 years. The PSM cohort included 10,830 women aged 65 and older. Of those, 7220 (66.6%) were identified as AOM never-users and 3610 (33.4%) as AOM ever-users. After PSM, no significant differences were observed in demographic and clinical features between the AOM groups, with all standardized mean difference values for covariates below 0.10, indicating an acceptable balance. Appendix A detail baseline characteristics according to the use of phentermine and liraglutide, respectively, before and after 1:2 PSM. The cohort for phentermine analysis comprised 6644 women aged ≥65 years, with 4430 (66.7%) identified as phentermine never-users and 2214 (33.3%) as phentermine ever-users. The liraglutide cohort included 807 women aged ≥65 years, of whom 538 (66.7%) were liraglutide never-users, and 269 (33.3%) were liraglutide ever-users. Following PSM, no significant differences were found in demographic and clinical features between the phentermine or liraglutide groups, with all standardized mean difference values for covariates remaining below 0.10, indicating an acceptable balance within each group.

Appendix A presents Kaplan–Meier estimates of time to incident ORCs, stratified by AOM use in a 1:2 propensity score-matched cohort. Median follow-up duration varied across cohorts: 49.2 months for ORCs, 55.2 months for BrCa, 62.3 months for CRC, 63.3 months for ECa, and 64.3 months for OCa. Overall survival rates for incident ORCs differed significantly between AOM ever-users and never-users (log-rank test, *p* < 0.001), with ever-users showing a survival advantage. Similar trends were observed in the incident BrCa (*p* < 0.001) and CRC (*p* = 0.010), but not in ECa (*p* = 0.086) or OCa (*p* = 0.065).

Figure 2, Figure 3 and Figure 4 display the findings of multivariable Cox proportional hazard models assessing associations between AOM, phentermine, and liraglutide use, and the risk of ORCs and cancer-specific sites (BrCa, CRC, ECa, and OCa) among women aged ≥65 years in a 1:2 propensity score-matched cohort. Of the 7220 AOM never-users, 2979 developed incident ORCs, whereas 1193 cases were identified among 3610 ever-users. Among the cancer subtypes, BrCa was the most common, with 1910 cases in never-users and 788 in ever-users, followed by CRC (594 vs. 226), ECa (329 vs. 129), and OCa (146 vs. 50). In the primary analysis (Figure 2), AOM use was associated with a significantly decreased risk of ORCs (adjusted hazard ratio [aHR]: 0.78; 95% confidence interval [CI]: 0.73–0.84; *p* < 0.001), advanced-stage ORCs (aHR: 0.76; 95% CI: 0.65–0.89; *p* < 0.001), BrCa (aHR: 0.84; 95% CI: 0.77–0.91; *p* < 0.001), CRC (aHR: 0.82; 95% CI: 0.71–0.96; *p* = 0.010), and advanced-stage CRC (aHR: 0.71; 95% CI: 0.54–0.91; *p* = 0.008). However, no significant association was observed between AOM use and the risk of ECa and OCa (Figure 2).

In the sensitivity analysis (Appendix A) for the primary analysis, the association between AOM use and the risk of ORCs was evaluated separately among women with and without diabetes. Among women with diabetes, AOM use was associated with a lower risk of several ORCs. Specifically, ever-use of AOM was associated with a 19% reduced risk of ORCs (aHR = 0.81; 95% CI: 0.73–0.89; *p* < 0.001). AOM use was also associated with a 34% reduction in advanced-stage ORCs (aHR = 0.66; 95% CI: 0.53–0.83; *p* < 0.001), a 23% lower risk of BrCa (aHR = 0.88; 95% CI: 0.78–0.99; *p* = 0.045), and a 23% reduced risk of CRC (aHR = 0.77; 95% CI: 0.62–0.95; *p* = 0.015). Additionally, AOM use was associated with a 34% reduction in advanced-stage CRC (aHR = 0.66; 95% CI: 0.46–0.94; *p* = 0.023) and a 51% lower risk of OCa (aHR = 0.49; 95% CI: 0.28–0.86; *p* = 0.012). Among women without diabetes, AOM use was only associated with a 22% lower risk of ORCs (aHR = 0.78; 95% CI: 0.71–0.85; *p* < 0.001) and a 19% reduced risk of BrCa (aHR = 0.81; 95% CI: 0.73–0.92; *p* < 0.001). Additionally, results from the competing risk models (Appendix A) were consistent with those from the conditional Cox models (Figure 2), supporting the robustness of our findings. The direction and magnitude of associations remained similar, indicating that accounting for the competing risk of death did not materially alter the observed relationships between AOM use and the risk of ORCs.

In the secondary analysis, phentermine use was associated with a significantly decreased risk of ORCs (aHR: 0.73; 95% CI: 0.67–0.79; *p* < 0.001), BrCa (aHR: 0.75; 95% CI: 0.67–0.84; *p* < 0.001), and ECa (aHR: 0.72; 95% CI: 0.54–0.97; *p* = 0.034). No significant association was observed between phentermine use and the risk of CRC and OCa (Figure 3). In the sensitivity analysis using competing risk models (Appendix A), results closely mirrored those from the conditional Cox models (Figure 3). The observed associations between AOM use and the risk of ORCs were stable across both approaches, suggesting that the risk of death had minimal impact on the observed relationships. Of note, among the 4430 phentermine never-users, 13 patients (0.3%) had prior exposure to other AOMs including liraglutide (*n* = 10), diethylpropion (*n* = 2), and orlistat (*n* = 1). The remaining 4417 patients had no exposure to any of the AOM examined in this study. This low level of prior AOM exposure is not expected to meaningfully influence the association between phentermine use and the risk of ORCs.

Lastly, liraglutide use was not associated with the risk of ORCs or its cancer-specific sites (Figure 4). Similar findings were observed in the sensitivity analysis with competing risk models (Appendix A) except for a significant association between liraglutide use and reduced risk of ORCs (sHR = 0.71; 95% CI: 0.54–0.93; *p* = 0.014.). However, these analyses are constrained by both a small sample size and a low incidence of ORCs in the liraglutide ever-user cohort, which together substantially reduce statistical power and increase the risk of a Type II error (false negative). As a result, the ability to detect even moderate associations is limited. While the absence of an association could suggest no clinically meaningful effect, we must be cautious in drawing definitive conclusions due to the insufficient sample size. Larger, well-powered studies are needed to more definitively assess its impact on the risk of ORCs. Additionally, among the 538 liraglutide never-users, 10 patients had prior exposure to phentermine, accounting for approximately 1.9% of the group. The remaining 528 patients had no exposure to any of the AOMs examined in this study. Given the small proportion of phentermine users, this is unlikely to have significantly impacted the observed findings.

## 4. Discussion

This population-based PSM cohort study investigated the association between the use of AOM and the risk of ORCs, including its cancer-specific sites BrCa, CRC, ECa, and OCa in older women. AOM use was associated with a decreased risk of ORCs, and these associations were more pronounced in older women with diabetes. When phentermine was evaluated separately as a primary exposure, we observed similar reduced risks for ORCs and BrCa, as well as for ECa in older women. This study is novel in quantifying the associations between AOMs, specifically phentermine and liraglutide, and the risk of ORCs and their cancer-specific sites. To our knowledge, this is the first population-based epidemiological study to examine these associations in older women.

Our findings resonate with a Canadian microsimulation study that underscored a potential link between second-generation AOMs, particularly GLP-1RAs, and a reduced risk of ORCs [29]. These results are consistent with earlier preclinical and clinical research, which demonstrated that GLP-1RAs can inhibit the growth of murine and human BrCa cell lines [30,31]. Additionally, Mendelian randomization studies indicate a lowered risk of BrCa associated with GLP-1RA use [32]. However, previous studies have found no significant association between AOM and BrCa risk [33,34,35].

Regarding CRC, our results are supported by an in vivo study showing that orlistat, an inhibitor of lipogenic enzymes, can suppress CRC development by reducing inflammation and inhibiting signal transducer and activator of transcription 3 (STAT3) and nuclear factor-kappa B (NF-κB) signaling pathways [36]. A large population-based cohort study also found that first- and second-generation GLP-1RAs, including liraglutide, semaglutide, exenatide, dulaglutide, and tirzepatide, were associated with a decreased CRC risk among individuals with obesity [37]. However, a separate large observational study did not find a significant association between orlistat and CRC risk [38].

In vitro studies suggest that orlistat may possess anti-tumor properties in both ECa and OCa [39,40]. Similarly, an in vivo study found that GLP-1RAs inhibited OCa cell growth, migration, and invasion, while promoting apoptosis by blocking the phosphatidylinositol 3-kinase/protein kinase B (PI3K/Akt) signaling pathway [41]. A recent retrospective cohort study further supported reduced risks of ECa and OCa with GLP-1RA use [35]. Despite this, our study did not observe statistically significant associations, either increased or decreased, between AOM use and the risk of ECa or OCa.

The association between AOM use and the risk of ORCs was directionally consistent across subgroups, with stronger and statistically significant effects observed in women with diabetes. In this group, AOM use was linked to reduced risks of overall ORCs, advanced-stage ORCs, BrCa, CRC, advanced-stage CRC, and OCa This pattern mirrors the primary analysis closely. Among women without diabetes, associations were limited to ORCs and BrCa. These findings suggest that the impact of AOMs on the risk of ORCs may be more evident among those with higher baseline metabolic risk.

In our secondary analysis, we examined phentermine and liraglutide separately to isolate the effects of each drug on the risk of ORCs. Phentermine is a primary amine that serves multiple roles as a dopaminergic, adrenergic, and sympathomimetic agent, functioning primarily as an appetite suppressant and central nervous system stimulant. It shares pharmacodynamic properties with amphetamine, as both act as trace amine-associated receptor 1 (TAAR1) agonists that facilitate neurotransmitter release [42]. Phentermine predominantly stimulates the release of norepinephrine, with lesser impacts on dopamine and serotonin. Additionally, it may trigger monoamine release via vesicular monoamine transporter 2 (VMAT2), similar to other substituted amphetamines [42,43]. Phentermine reduces hunger by targeting specific hypothalamic nuclei and enhances fat breakdown in adipose tissue through elevated norepinephrine and epinephrine levels [44]. Our study demonstrated that phentermine use is associated with reduced risk of ORCs, BrCa, and ECa. To our knowledge, no prior studies have evaluated the independent association between phentermine use and ORC risk in older women. This paucity of research limits the direct comparison of our findings with the existing literature.

Liraglutide, a synthetic GLP-1 RA, mimics the actions of the natural GLP-1 hormone. It regulates blood sugar levels and influences appetite by acting on both peripheral tissues, like the pancreas and stomach, as well as central nervous system pathways in the brain [45]. In vitro study has shown that liraglutide exhibits anti-proliferative effects on Michigan Cancer Foundation-7 (MCF-7) BrCa cells when cultured in media derived from obese adipose tissue-derived stem cells (ADSCs-CM), which release cytokines and growth factors that encourage cancer growth and metastasis, and can effectively reduce inflammatory mediators, mitigate the pro-proliferative effects of leptin, and amplify the anti-proliferative impact of adiponectin [46]. Another in vitro study demonstrated that liraglutide decreases BrCa cell proliferation and promotes apoptosis in a dose-dependent manner by inhibiting miRNA-27a, which subsequently upregulates adenosine monophosphate-activated protein kinase (AMPKα2) expression [47]. Despite these promising findings, our analysis found no significant association between liraglutide use and the risk of ORCs, which may be attributable to the small number of liraglutide users in our cohort.

Collectively, our findings, supported by preclinical and clinical studies, suggest that AOM use may be associated with a reduced risk of BrCa, CRC, and ECa through various mechanisms. One potential key mechanism is the reduction of adipose tissue. This reduction subsequently decreases the production of pro-inflammatory cytokines and hormones, such as estrogen, leptin, and insulin, all of which are implicated in carcinogenesis. A recent systematic review and meta-analysis found that weight-loss interventions, including pharmacotherapy, significantly reduced the levels of inflammatory biomarkers, leptin, and estrogen hormones associated with the reduced risk of ECa [48]. Additionally, weight loss has been shown to alter the adipose tissue microenvironment and systemic endocrine profile in ways that inhibit tumor development and progression [49]. However, this potential mechanism remains speculative in our study due to the lack of BMI measure in our data, which prevents a direct assessment of weight change in relation to cancer risk. Therefore, any conclusions regarding the effect of weight loss on the risk ORCs in our cohort must be interpreted with caution.

A key strength of our study lies in the use of the SEER-Medicare dataset, which provides a diverse and geographically representative sample of the U.S. population [21,22]. We effectively managed immortal time bias by ensuring that both exposed and unexposed groups maintained continuous Medicare Parts A and B enrollment for at least six months before the first exposure date, and this date was aligned in the matched cohort and at least six months before the diagnosis of ORCs (if any) [50]. However, several limitations must be acknowledged. The retrospective nature of the SEER-Medicare data inherently carries risks of selection and measurement biases that may significantly impact the reproducibility and comparability of studies [21,22]. While we employed 1:2 PSM to address potential selection bias, this method can only account for measured confounders [26,27]. Unmeasured confounding remains a concern, particularly from factors such as genetic predispositions and lifestyle behaviors, including physical activity, smoking, and alcohol consumption. The lack of reliable data on BMI (≥30 kg/m^2^), a standard indicator of obesity, further contributes to this limitation. Although SEER-Medicare includes BMI data, it is often underreported and inconsistently coded, which undermines its completeness and validity. While the SEER-Medicare does not explicitly advise against the use of BMI, it flags the variable as problematic due to its potential to introduce misclassification bias and lead to inaccurate adjustments [51]. To address this limitation, we adjusted for several well-established proxies of obesity, including diabetes, hypertension, hyperlipidemia, insulin use, and CCI. These covariates likely captured a meaningful proportion of obesity-related risk. Nevertheless, the possibility of residual confounding due to unmeasured or incompletely measured variables cannot be fully excluded. The six-month pre-index period may not fully capture the complexity of a patient’s medical history. While the data analyzed in this study predate the recent availability and widespread use of new AOMs, we acknowledge that these developments may influence current trends in weight management. However, the impact of these newer treatments on our findings is beyond the scope of this analysis and warrants further investigation in future studies. Although we confirmed that no patients had concurrent use of the AOMs included in this study (diethylpropion, liraglutide, lorcaserin, orlistat, phendimetrazine, and phentermine) during the exposure assessment period (July 2007–June 2015), we were unable to assess AOM use prior to 2007 due to the absence of Medicare Part D data [21]. Consequently, a formal wash-out period could not be implemented, which may have resulted in the inadvertent inclusion of some prevalent AOM users, despite design measures aimed at mitigating this risk. In addition, we did not capture exposure to AOM outside those specified such as other GLP-1 RAs which may have influenced the observed associations. Furthermore, screening bias may have impacted the detection rates of ORCs, particularly among individuals with lower socioeconomic status (SES), who often experience delayed or limited access to cancer screening. These disparities may have led to underestimation of ORCs incidence. Although the dataset did not include direct measures of cancer screening practices, we adjusted for SES-related variables such as race/ethnicity, comorbidities, and poverty status that serve as commonly accepted proxies in cancer epidemiology. Yet, residual confounding remains. While the factors we adjusted for may not fully reflect individual-level screening behaviors, they offer a pragmatic and widely used approach to account for socioeconomic disparities in access to preventive care. Additionally, the lack of data on medication dosage and duration precludes investigation of time– and dose–response associations. Finally, our study’s focus on older women with Medicare coverage restricts the generalizability of our findings to men, younger women, individuals with alternative insurance statuses, or those who are uninsured. Given these limitations, our conclusions should be interpreted with caution.

## 5. Conclusions

In summary, this study offers insights into the association between AOM use and the risk of ORCs in older women using PSM-matched cohort analyses. Our primary analysis demonstrated a significant association between AOM use and the reduced risk of ORCs and its subgroups, including advanced-stage ORCs, BrCa, CRC, and advanced-stage CRC. The association was more pronounced in women with diabetes, suggesting that the observed survival benefit may be greater among those with higher baseline risk. Our secondary analysis found that phentermine use was significantly associated with a reduced risk of ORCs and its subgroups, including BrCa, and ECa. In contrast, no significant association was observed between liraglutide and the risk of ORCs or its subgroups. Despite the limitations inherent to retrospective studies, these findings provide preliminary evidence for the potential protective effects of AOMs against ORCs in older women. The compelling findings of phentermine use are of paramount significance and could provide justification for clinical trials aimed at validating their efficacy as potential BrCa- and ECa-preventive strategies in older women with risk factors such as obesity, genetic alterations (e.g., BRCA1/2, Lynch syndrome), family history of cancer, and hormone replacement therapy use. Further prospective studies with extended follow-up are warranted to validate these findings, investigate the underlying mechanisms of action, and inform clinical decision-making.

## Figures and Tables

**Figure 1 cancers-17-01624-f001:**
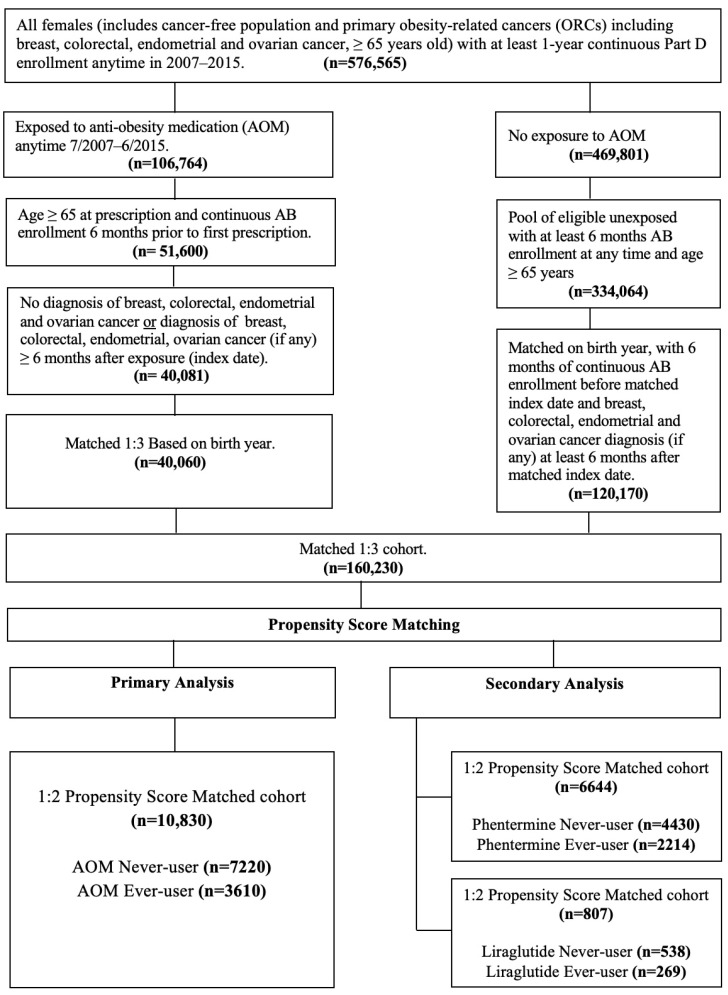
Cohort selection flowchart. Study inclusion and exclusion criteria.

**Figure 2 cancers-17-01624-f002:**
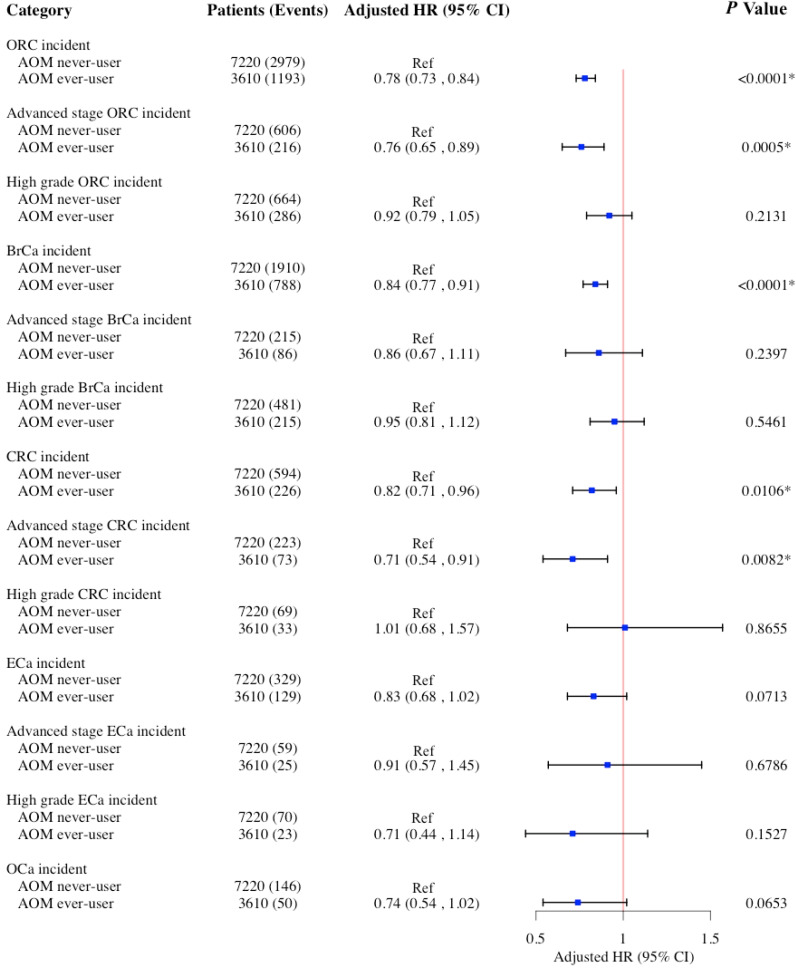
HRs and 95% CIs for the association between the use of AOM and the risk of ORCs and its subgroups (BrCa, CRC, ECa, and OCa) among women ≥65 years old in 1:2 propensity score-matched cohort. Cox models adjusted for age at index date after propensity score matching. AOMs include diethylpropion, liraglutide, lorcaserin, orlistat, phendimetrazine, and phentermine. * Statistical significance at the *p* value < 0.05 level. Abbreviations: aHR, adjusted hazard ratio; AOM, anti-obesity medication; BrCa, breast cancer; CI, confidence interval; CRC, colorectal cancer; ECa, endometrial cancer; OCa, ovarian cancer; ORCs, obesity-related cancers.

**Figure 3 cancers-17-01624-f003:**
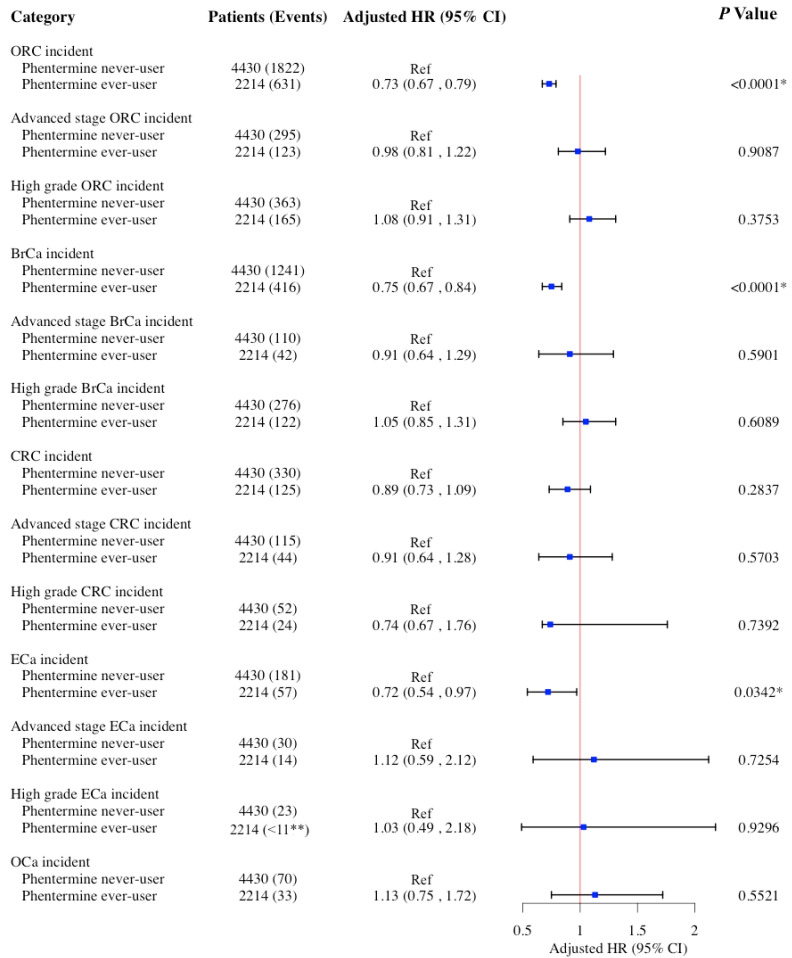
HRs and 95% CIs for the association between the use of phentermine and risk of ORCs and its subgroups (BrCa, CRC, ECa, and OCa) among women ≥65 years old in 1:2 propensity score-matched cohort. Cox models adjusted for age at index date after propensity score matching. * Statistical significance at the *p* value < 0.05 level. ** SEER-Medicare data presentation guideline has been followed and all counts less than 11 have been suppressed. Abbreviations: aHR, adjusted hazard ratio; BrCa, breast cancer; CI, confidence interval; CRC, colorectal cancer; ECa, endometrial cancer; OCa, ovarian cancer; ORCs, obesity-related cancers.

**Figure 4 cancers-17-01624-f004:**
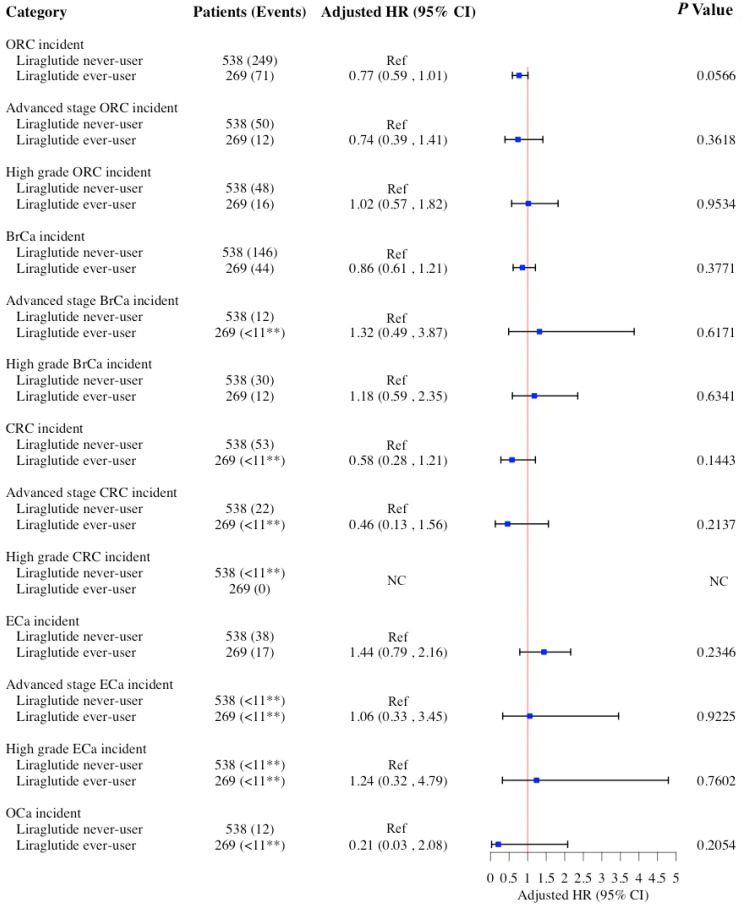
HRs and 95% CIs for the association between use of liraglutide and risk of ORCs and its subgroups (BrCa, CRC, ECa, and OCa) among women ≥65 years old in 1:2 propensity score-matched cohort. Cox models adjusted for age at index date after propensity score matching. ** SEER-Medicare data presentation guideline has been followed and all counts less than 11 have been suppressed. Abbreviations: aHR, adjusted hazard ratio; BrCa, breast cancer; CI, confidence interval; CRC, colorectal cancer; ECa, endometrial cancer; NC, not calculated; OCa, ovarian cancer; ORCs, obesity-related cancers.

**Table 1 cancers-17-01624-t001:** Baseline characteristics according to the use of AOM before and after 1:2 propensity score matching.

Characteristics	No. (%)							
	Pre- 1:2 Propensity Score Matching (*n* = 160,230)				Post- 1:2 Propensity Score Matching (*n* = 10,830)			
	AOM Never-User 156,619 (97.7)	AOM Ever-User 3611 (2.3)	*p*-Value	% Std.Diff	AOM Never-User 7220 (66.6)	AOM Ever-User 3610 (33.4)	*p*-Value	% Std.diff
Age at index date			<0.0001 *	0.3206			0.8277	0.0413
65–70	16,907 (10.8)	605 (16.8)			1210 (16.8)	605 (16.8)		
70–75	38,488 (24.6)	1208 (33.5)			2421 (33.5)	1208 (33.5)		
75–80	39,166 (25.0)	882 (24.4)			1761 (24.4)	882 (24.4)		
≥80	62,058 (39.6)	619 (25.3)			1827 (25.3)	915 (25.3)		
Race/Ethnicity			<0.0001 *	0.1377			0.6006	0.0546
White	123,298 (78.7)	2967 (82.2)			5949 (82.4)	2966 (82.2)		
Black	15,398 (9.8)	336 (9.3)			673 (9.3)	336 (9.3)		
Hispanic	4825 (3.1)	131 (3.63)			250 (3.5)	131 (3.6)		
Other	13,098 (8.4)	177 (4.9)			347 (4.8)	177 (4.9)		
CCI			<0.0001 *	0.4974			0.9545	0.0303
0	77,196 (49.3)	958 (26.5)			1906 (26.4)	958 (26.4)		
1	44,705 (28.5)	1231 (34.1)			2464 (34.1)	1231 (34.1)		
2	20,092 (12.8)	746 (20.7)			1480 (20.5)	746 (20.6)		
3 or more	14,626 (9.4)	676 (18.7)			1369 (18.9)	675 (18.7)		
Diabetes	46,668 (29.8)	1656 (45.9)	<0.0001 *	0.3359	3327 (46.1)	1656 (45.9)	0.5763	−0.0114
Use of insulin	7462 (4.76)	421 (11.7)	<0.0001 *	0.2531	946 (11.7)	421 (11.7)	0.8152	0.0048
Use of metformin	36,449 (23.3)	1383 (38.3)	<0.0001 *	0.3206	2831 (39.2)	1383 (38.3)	0.3650	−0.0185
Hypertension	94,951 (60.6)	2636 (73.0)	<0.0001 *	0.2651	5287 (73.2)	2635 (73.0)	0.7242	−0.0072
Hyperlipidemia	66,442 (42.4)	1936 (53.6)	<0.0001 *	0.2254	3828 (53.0)	1935 (53.1)	0.5953	0.0108
CVD	75,677 (48.3)	2205 (61.1)	<0.0001 *	0.2581	4429 (61.4)	2204 (61.1)	0.6252	−0.0100
Malaise and fatigue	24,529 (15.7)	976 (27.0)	<0.0001 *	0.2801	1927 (26.7)	975 (27.0)	0.9390	0.0016
Muscular wasting	1530 (0.9)	100 (2.8)	<0.0001 *	0.1325	184 (2.6)	100 (2.8)	0.2802	0.0218
Hypogonadism ^†^	28 (0.1)	0 (0.0)	1.0000	−0.0189	<11 (<0.1) ^‡^	0 (0.0)	0.5555	−0.0288
APD ^†^	93 (0.1)	<11 (<0.2) ^‡^	0.1760	0.0176	<11 (<0.1) ^‡^	<11 (<0.2) ^‡^	1.0000	−0.0040
Depression disorder	8784 (5.6)	506 (14.0)	<0.0001 *	0.2854	999 (13.8)	505 (13.9)	0.9063	0.0024
Osteoporosis	20,379 (13.0)	552 (15.3)	<0.0001 *	0.0653	1100 (15.2)	552 (15.3)	0.8055	0.0050
Cushing’s syndrome	40 (0.1)	<11 (<0.1) ^‡^	0.0726	0.0247	<11 (<0.1) ^‡^	<11 (<0.1) ^‡^	1.0000	−0.0046
Hypothyroidism	28,878 (18.4)	1023 (28.3)	<0.0001 *	0.2353	2052 (28.2)	1023 (28.3)	0.8566	−0.0037
Hyperthyroidism	1797 (1.15)	53 (1.5)	0.0748	0.0282	100 (1.4)	53 (1.5)	0.6591	−0.0090
PCOS ^†^	19 (0.1)	<11 (<0.1) ^‡^	0.3661	0.0110	<11 (<0.1) ^‡^	<11 (<0.1) ^‡^	1.0000	0.0000
Poverty, mean (SD) ^§^	11.5 (8.8)	11.8 (8.7)	0.0271 *	0.0373	12.0 (8.9)	11.9 (8.7)	0.2434	−0.0237

Abbreviations: APD, anterior pituitary dysfunction; CVD, cardiovascular disease; CCI, Charlson comorbidity index; PCOS, polycystic ovary syndrome; SD, standard deviation; std.diff, standardized difference. * Chi-square statistical significance at the *p* value < 0.05 level. ^†^ Fisher exact test two-sided *p*-value. ^‡^ SEER-Medicare data presentation guideline has been followed and all counts less than 11 have been suppressed. ^§^ Percent of residents living below poverty.

## Data Availability

The SEER-Medicare datasets used to conduct this study are available upon approval of a research protocol by the National Cancer Institute. Instructions for obtaining these data are available at https://healthcaredelivery.cancer.gov/seermedicare/obtain/ (accessed on 11 April 2024). This study used the linked SEER-Medicare database. The interpretation and reporting of these data are the sole responsibility of the authors.

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
