# Peer review of "Anti-Obesity Medications and the Risk of Obesity-Related Cancers in Older Women: A Propensity Score Matching Analysis of 2007–2015 SEER-Medicare Data"

_cancers, 2025, doi:10.3390/cancers17101624_

Round 1
Reviewer 1 Report
Comments and Suggestions for Authors
Anti-obesity medications and risk of obesity-related cancers in older women: A propensity score matching analysis of 2007– 2015 SEER-Medicare
The author in the article studied and investigated the link between anti-obesity medications (AOMs) and obesity related cancers (ORCs) in women aged 65 years and older. They performed a retrospective study on SEER-Medicare data from 2007 – 2015 and found that anti-obesity medicines were associated with significantly lower risk of obesity related cancers. This study is interesting as it suggests a potential protective effects of AOMs against ORCs in older women. However, I have following points mentioned below which hinders my enthusiasm for this manuscript.
1). In the introduction section of the article, the author names several AOMs but limited details regarding them. A one or two sentences explaining how these drugs differ will enhance the context.
2). Given that this is a retrospective study, it is understandable why the author did not the current GLP-1Ra’s in the introduction. However, I feel referencing incretin-based therapies which are common weight loss drug today would enhance the background and provides reader with a more comprehensive background of the topic.
3). In the line 72-74, the author cites stats from the American Cancer society immediately after stating that “ORCs are increasing at alarming rates”. This may unintentionally imply that the cited stats mentioned reflects only ORCs, when in fact they represent the total number of cases and not necessarily all due to ORCs. To avoid this misleading association the author should rephrase this sentence for clarity.
4). The material and method section of the study is well written, well-structured, and detailed. However, there are few formatting issues like overuse of passive voice, redundancy in subsections. These need to be fixed.
5). A small point: the way the material and method section currently structured provides ambiguity in index date and exposure classification specially for the unexposed individuals. Clarifying how the index date was assigned to unexposed individuals would improve understanding.
6). The author fails to mention if time-varying confounding factors like duration of AOM use, cancer screening practices, weight changes was considered as these could influence ORCs risk.
7). The author mentioned the outcomes of ORCs in section 2.4, but the biomarkers for these outcomes are not mentioned in the material and method section which should be mentioned for ease to follow.
8). I really like the way the author summarizes the results. It is clear and easy to follow.
9). I am concerned about how the authors have presented the liraglutide analysis. Firstly, the small sample size of the liraglutide cohort is not acknowledged, which is a critical limitation that should be addressed. The authors should provide a rationale or discussion regarding this constraint. Secondly, the results for liraglutide are summarized in only two sentences, lacking sufficient context or interpretation. A more thorough explanation of the null findings, including potential limitations such as limited statistical power, would greatly strengthen the credibility and completeness of the analysis.
10). The manuscript does not provide information on the number of AOMs used per patient, particularly in the secondary analyses involving liraglutide and phentermine. Given the availability of comprehensive data, it is unclear why the potential impact of dual or multiple AOM use on ORC outcomes has not been considered. I recommend that the authors either conduct additional analyses to account for concurrent AOM use or clarify whether patients were exclusive users of a single AOM. This information is essential to better understand the observed associations, especially regarding phentermine’s effects on ORCs, BrCa, and ECa.
Author Response
Comments 1: In the introduction section of the article, the author names several AOMs but limited details regarding them. A one or two sentences explaining how these drugs differ will enhance the context.
Response 1: Thank you for your suggestion. To enhance the context in the Introduction section, we have revised the manuscript as follows:
(page 2, lines 33-44) “Earlier AOMs such as phentermine, liraglutide, diethylpropion, lorcaserin, orlistat, and phendimetrazine have gained popularity in obesity management, mainly due to their low cost [13]. These medications act through distinct mechanisms. Phentermine, diethylpropion, and phendimetrazine function as sympathomimetic appetite suppressants. Orlistat works by inhibiting fat absorption in the gastrointestinal tract. Liraglutide, a glucagon-like peptide-1 receptor agonist (GLP-1 RA), helps regulate appetite and food intake, while lorcaserin targets serotonin 2C receptors to promote satiety [14]. Their effectiveness in promoting weight loss is well documented.”
We have also provided context by recognizing the use of earlier AOM as mentioned in the Discussion section of the original manuscript:
page 14, lines 3-7 “While the data analyzed in this study predate the recent availability and widespread use of new AOM, we acknowledge that these developments may influence current trends in weight management. However, the impact of these newer treatments on our findings is beyond the scope of this analysis and warrants further investigation in future studies.”
Comments 2: Given that this is a retrospective study, it is understandable why the author did not the current GLP-1Ra’s in the introduction. However, I feel referencing incretin-based therapies which are common weight loss drug today would enhance the background and provides reader with a more comprehensive background of the topic.
Response 2: Thank you for this thoughtful comment. We have now included a brief mention of these therapies in the Introduction section to acknowledge their growing relevance in obesity management:
(page 2, lines 40-43) “While our study focuses on earlier AOM, it is important to acknowledge the growing use of newer GLP-1 RA therapies, such as semaglutide and tirzepatide. These agents have demonstrated substantial efficacy in promoting weight loss and improving metabolic control, and they are reshaping current approaches to obesity treatment [15].”
Comments 3: In the line 72-74, the author cites stats from the American Cancer society immediately after stating that “ORCs are increasing at alarming rates”. This may unintentionally imply that the cited stats mentioned reflects only ORCs, when in fact they represent the total number of cases and not necessarily all due to ORCs. To avoid this misleading association the author should rephrase this sentence for clarity.
Response 3: Thank you for pointing this out. We agree that the original phrasing may have unintentionally implied that the cited cancer statistics refer exclusively to ORCs. To address this concern, we have revised the Introduction section as follows:
(page 2, lines 24-30) “The prevalence of obesity has reached alarming levels in the United States, affecting 42% of adults. Moreover, the age-adjusted prevalence of severe obesity (a body mass index [BMI] of 40 kg/m2 or higher) is 9.2% and is higher in women than in men [10]. According to the American Cancer Society’s 2025 report, an estimated 316,950 new cases of BrCa, 71,810 of CRC, 69,120 of ECa, and 20,890 of OCa were expected in women [11]. While not all cases are attributable to obesity, these cancers have been associated with excess body weight in the literature [7-9].”)
Comments 4: The material and method section of the study is well written, well-structured, and detailed. However, there are few formatting issues like overuse of passive voice, redundancy in subsections. These need to be fixed.
Response 4: Thank you for your positive feedback and helpful suggestions. In response, we have carefully revised the Materials and Methods section to reduce the use of passive voice, eliminate redundancies within subsections, and improve overall clarity and flow (from page 3 to page 6).
Comments 5: A small point: the way the material and method section currently structured provides ambiguity in index date and exposure classification specially for the unexposed individuals. Clarifying how the index date was assigned to unexposed individuals would improve understanding.
Response 5: Thank you for your helpful comment. We agree that the original description of index date assignment for unexposed individuals could be clearer. To address this, we have revised the Study Population subsection in the Methods section as follows:
(page 3, lines 23-31) “Patients were categorized as AOM-exposed or unexposed based on prescription data within this period. The index date for exposed individuals was the first AOM prescription date. We excluded those <65 years at index date, with <6 months of Medicare Parts A and B coverage prior to index date or diagnosed with ORCs (BrCa, CRC, ECa, OCa) within six months of index date. We matched each exposed patient to three unexposed patients (1:3) by birth year. For unexposed individuals, the index date was assigned based on the index date of their matched exposed counterpart to ensure comparable follow-up timeframes (matched cohort n = 160,230).”
Comments 6: The author fails to mention if time-varying confounding factors like duration of AOM use, cancer screening practices, weight changes was considered as these could influence ORCs risk.
Response 6: We appreciate your important observation regarding time-varying confounding. The duration of AOM use was neither approved nor available in our SEER-Medicare analytic dataset, which precludes an accurate investigation of the time-response association. We have acknowledged this limitation in the Discussion section of the original manuscript: (page 14, lines 24-25): “Additionally, the lack of data on medication dosage and duration precludes investigation of time- and dose-response associations.”
Similar to AOM use duration, direct measures of cancer screening behavior were neither approved nor available in our SEER-Medicare analytic dataset. However, we accounted for potential confounding by adjusting for variables commonly used as proxies for socioeconomic status, such as race/ethnicity, comorbid conditions, and poverty status. These factors are strongly associated with differential access to preventive care, including cancer screening. Yet we acknowledge that residual confounding remains. Although we recognize the limitations of these proxy variables, they offer a reasonable strategy to partially mitigate potential screening-related bias in observational cancer epidemiology. We revised the Discussion section to include this limitation as follows:
(page 14, lines 15–24) “Furthermore, screening bias may have impacted the detection rates of ORCs, particularly among individuals with lower socioeconomic status (SES), who often experience delayed or limited access to cancer screening. These disparities may have led to underestimation of ORCs incidence. Although the dataset did not include direct measures of cancer screening practices, we adjusted for SES-related variables such as race/ethnicity, comorbidities, and poverty status that serve as commonly accepted proxies in cancer epidemiology. While these factors may not fully reflect individual-level screening behaviors, they offer a pragmatic and widely used approach to account for socioeconomic disparities in access to preventive care.”
Regarding weight changes, our 2007-2015 SEER-Medicare dataset does not include information on BMI, or any other biological markers related to obesity (e.g., laboratory data). Although SEER-Medicare includes ICD-9 and ICD-10 codes (in effect as of 10/15/15) that can define obesity, it does not recommend using these variables due to their low sensitivity. We revised the Discussion section to reflect this limitation as follows:
(page 13 line 44 to page 14 line 2) “The lack of reliable data on BMI (≥30 kg/m²), a standard indicator of obesity, further con-tributes to this limitation. Although SEER-Medicare includes BMI data, it is often underreported and inconsistently coded, which undermines its completeness and validity. While the SEER-Medicare does not explicitly advise against the use of BMI, it flags the variable as problematic due to its potential to introduce misclassification bias and lead to inaccurate adjustments [53]. To address this limitation, we adjusted for several well-established proxies of obesity, including diabetes, hypertension, hyperlipidemia, insulin use, and CCI. These covariates likely captured a meaningful proportion of obesity-related risk. Nevertheless, the possibility of residual confounding due to unmeasured or incompletely measured variables cannot be fully excluded.”
Comments 7: The author mentioned the outcomes of ORCs in section 2.4, but the biomarkers for these outcomes are not mentioned in the material and method section which should be mentioned for ease to follow.
Response 7: We appreciate the feedback and have revised the ORCs Outcomes subsection in the Methods section for improved clarity as follow:
(page 4 line 4 to page 5 line 9 145-163) “We examined the first incident ORCs, advanced stage ORCs, high grade ORCs, BrCa, advanced stage BrCa, high grade BrCa, CRC, advanced stage CRC, high grade CRC, ECa, advanced stage ECa, high grade ECa, and OCa. These cancers were selected based on consistent associations with excess body fat identified by the International Agency for Re-search on Cancer (IARC) [25]. Incident ORCs was identified using International Classification of Diseases for Oncology-3 (ICD-O-3) site and histology codes from the SEER data-base. BrCa was defined by site codes C50.0–C50.9 and histology codes 8500–8543. CRC included site codes C18.0–C20.9 and histology codes 8140–8147, 8210–8211, 8220–8221, 8260–8263, 8480–8481, and 8490. ECa was defined by site codes C54.0–C55.9 and histology codes 8140, 8262, 8323, 8380–8384, 8480–8482, and 8560. OCa was identified using site code C56.9 and histology codes 8440–8469. Cancer staging was based on the Ameri-can Joint Committee on Cancer (AJCC) system available in SEER. We assigned the first day of the diagnosis month to enhance censoring accuracy, as SEER reports only the month and year of cancer diagnosis. Time-to-event was calculated in months starting from the index date (first AOM exposure). For patients who developed an ORCs, time-to-event was measured from the index date to the first ORCs diagnosis. Patients who did not develop ORCs were right-censored at the study end date (December 31, 2015) or at the time of loss to follow-up, whichever occurred first. Patients who remained ORCs-free and died before the study end date were right-censored at the time of death.
Comments 8: I really like the way the author summarizes the results. It is clear and easy to follow.
Response 8: Thank you for your kind feedback. We are pleased to hear that you found the results section clear and easy to follow.
Comments 9: I am concerned about how the authors have presented the liraglutide analysis. Firstly, the small sample size of the liraglutide cohort is not acknowledged, which is a critical limitation that should be addressed. The authors should provide a rationale or discussion regarding this constraint. Secondly, the results for liraglutide are summarized in only two sentences, lacking sufficient context or interpretation. A more thorough explanation of the null findings, including potential limitations such as limited statistical power, would greatly strengthen the credibility and completeness of the analysis.
Response 9: Thank you for highlighting this important limitation. We have revised the Results section as follows:
(page 10 line 12 to page 11 line 8) “Similar findings were observed in the sensitivity analysis with competing risk models (Supplementary Table S7) except for a significant association between liraglutide use and reduced risk of ORCs (sHR = 0.71; 95% CI: 0.54–0.93; p = 0.014.). However, these analyses are constrained by both a small sample size and a low incidence of ORCs in the liraglutide ever-user cohort, which together substantially reduce statistical power and increase the risk of a Type II error (false negative). As a result, the ability to detect even moderate associations is limited. While the absence of an association could suggest no clinically meaningful effect, we must be cautious in drawing definitive conclusions due to the insufficient sample size. A larger, well-powered studies are needed to more definitively assess its impact on ORCs risk. Additionally, among the 538 liraglutide never-users, 10 patients had prior exposure to phentermine, accounting for approximately 1.9% of the group. The remaining 528 patients had no exposure to any of the AOM examined in this study. Given the small proportion of phentermine users, this is unlikely to have significantly impacted the observed findings.”
Comments 10: The manuscript does not provide information on the number of AOMs used per patient, particularly in the secondary analyses involving liraglutide and phentermine. Given the availability of comprehensive data, it is unclear why the potential impact of dual or multiple AOM use on ORC outcomes has not been considered. I recommend that the authors either conduct additional analyses to account for concurrent AOM use or clarify whether patients were exclusive users of a single AOM. This information is essential to better understand the observed associations, especially regarding phentermine’s effects on ORCs, BrCa, and ECa.
Response 10: We appreciate your diligence in pointing out this potential limitation. We conducted a thorough review and confirmed that no patients had concurrent use of the AOMs examined in this study. The number of phentermine and liraglutide users is presented in Figure 1, and descriptive analyses of the secondary analysis cohorts, both before and after propensity score matching, are provided in the supplementary materials (Supplementary Tables S1 and S2). We have revised the Discussion section to reflect this information as follows:
(page 14, lines 7-15) “Although we confirmed that no patients had concurrent use of the AOMs included in this study (diethylpropion, liraglutide, lorcaserin, orlistat, phendimetrazine, and phentermine) during the exposure assessment period (July 2007–June 2015), we were unable to assess AOM use prior to 2007 due to the absence of Medicare Part D data [23]. Consequently, a formal wash-out period could not be implemented, which may have resulted in the inadvertent inclusion of some prevalent AOM users, despite design measures aimed at mitigating this risk. In addition, we did not capture exposure to AOM outside those specified such as other GLP-1 RAs which may have influenced the observed associations.”
We also added information regarding the comparison group (never-user) in the Results section for transparency as follows:
Page 9, lines 17-22 “Of note, among the 4,430 phentermine never-users, 13 patients (0.3%) had prior exposure to other AOMs including liraglutide (n = 10), diethylpropion (n = 2), and orlistat (n = 1). The remaining 4,417 patients had no exposure to any of the AOM examined in this study. This low level of prior AOM exposure is not expected to meaningfully influence the association between phentermine use and ORCs risk.”
page 11, lines 4-8 “Additionally, among the 538 liraglutide never-users, 10 patients had prior exposure to phentermine, accounting for approximately 1.9% of the group. The remaining 528 patients had no exposure to any of the AOM examined in this study. Given the small proportion of phentermine users, this is unlikely to have significantly impacted the observed findings.”
Again, we thank you very much for the thoughtful and constructive comments, which have significantly improved the quality of our manuscript. We believe that the revisions we have made address your concerns and strengthen the overall clarity and rigor of the study.
Reviewer 2 Report
Comments and Suggestions for Authors
The study utilized SEER-Medicare database to explore the impact of anti-obesity medications (AOM) on cancer development.
- Did authors apply some wash-out period to exclude prevalent users?
- For phentermine and liraglutide cohorts, were there any patients exposed to both agents? If so, how did authors handle these patients?
- Some patients might only receive just one claim of AOC of interest, it would be helpful if the authors could conduct some sensitivity analyses taking length of exposure into consideration.
- What were the time frames to define clinic related features?
- How did the authors handle conditions, such as diabetes, CVD, which were listed separately in the clinic related features but also included in the calculation of Charlson Comorbidity Index?
-
Death could be a competing risk.
- For cancer-specific analyses, if a patient developed BrCa first then CRC, would the patient only be included in the BrCa related analyses or in both BrCa and CRC? If former, other ORC except of the cancer of interest should also be considered as competing risks.
- One major concern is the underlying differences between exposed and non-exposed patients. Exposed patients were overweight or obese, yet unexposed patients might be normal weight or even under-weight. Although authors have applied propensity score matching to balance the potential confounding between the two groups, it might be helpful if the authors could conduct some sensitivity analyses among more homogeneous population, for example patients with diabetes, and add more discussions on it.
-
Minor comment: The title for 2.7 is wrong.
Author Response
Comments 1: Did authors apply some wash-out period to exclude prevalent users?
Response 1: Thank you for your insightful question. We did not implement a formal wash-out period prior to July 2007 due to the unavailability of Medicare Part D data before that time. However, we required at least six months of continuous Medicare Parts A and B enrollment before the index date, and AOM exposure was identified from the earliest possible time point (July 2007) onward. These design choices helped reduce the inclusion of prevalent users to the extent possible within the constraints of the data. We have acknowledged this potential limitation in the revised Discussion section as follows:
(page 14, line 7-14) “Although we confirmed that no patients had concurrent use of the AOMs included in this study (diethylpropion, liraglutide, lorcaserin, orlistat, phendimetrazine, and phentermine) during the exposure assessment period (July 2007–June 2015), we were unable to assess AOM use prior to 2007 due to the absence of Medicare Part D data [23]. Consequently, a formal wash-out period could not be implemented, which may have resulted in the inadvertent inclusion of some prevalent AOM users, despite design measures aimed at mitigating this risk.”
Comments 2: For phentermine and liraglutide cohorts, were there any patients exposed to both agents? If so, how did authors handle these patients?
Response 2: Thank you for raising this important point. We conducted a thorough review and confirmed that patients classified as exposed in the phentermine and liraglutide cohorts were only exposed to a single agent. We also added information regarding the comparison group (never-user) in the Results section for transparency as follows:
(page 9, lines 17-22) “Of note, among the 4,430 phentermine never-users, 13 patients (0.3%) had prior exposure to other AOMs including liraglutide (n = 10), diethylpropion (n = 2), and orlistat (n = 1). The remaining 4,417 patients had no exposure to any of the AOM examined in this study. This low level of prior AOM exposure is not expected to meaningfully influence the association between phentermine use and ORCs risk.”
(page 11 , lines 4-8) “Additionally, among the 538 liraglutide never-users, 10 patients had prior exposure to phentermine, accounting for approximately 1.9% of the group. The remaining 528 patients had no exposure to any of the AOM examined in this study. Given the small proportion of phentermine users, this is unlikely to have significantly impacted the observed findings.”
Comments 3: Some patients might only receive just one claim of AOC of interest, it would be helpful if the authors could conduct some sensitivity analyses taking length of exposure into consideration.
Response 3: We appreciate your thoughtful suggestion. As mentioned in the Discussion section (page 14, lines 24-25), “Additionally, the lack of data on medication dosage and duration precludes investigation of time- and dose-response associations.” Unfortunately, due to this limitation, we were unable to conduct sensitivity analyses that would account for the length of exposure. We hope this clarifies the constraints of our analysis, and we appreciate your understanding.
Comments 4: What were the time frames to define clinic related features?
Response 4: Thank you for highlighting this point. As stated on (page 5, lines 18-20) "All covariates were measured at least six months before the index date and the mandatory six-month gap between medication initiation and ORCs diagnosis date to mitigate ascertainment bias."
Comments 5: How did the authors handle conditions, such as diabetes, CVD, which were listed separately in the clinic related features but also included in the calculation of Charlson Comorbidity Index?
Response 5: Thank you for this important question. Prior to propensity score matching, we evaluated multicollinearity during covariate selection using the Variance Inflation Factor (with <5 cut-off) and found no evidence of concern among the included predictors. Accordingly, diabetes and CVD were retained both as individual covariates and as components of the Charlson Comorbidity Index in the logistic regression model used to estimate propensity scores. Following matching, the cohorts were well balanced on these and other variables; thus, we did not re-adjust for them in the conditional Cox models. This approach allowed us to account for both overall comorbidity burden and the independent effects of key conditions, without redundancy or model overfitting. We have added a new Supplementary Table S1 in supplementary materials for multicollinearity analysis and revised the Covariates subsection in the Methods section as follows:
(page 5, lines 11-18) “Covariates included demographic and clinical characteristics, selected based a priori knowledge [26]. Demographics included age at index date, race/ethnicity, and poverty status. Clinical variables included the NCI-Charlson Comorbidity Index (CCI) [27], diabetes, insulin and metformin use, hyperlipidemia, cardiovascular disease, malaise/fatigue, muscular wasting, hypogonadism, anterior pituitary dysfunction, depression, osteoporosis, Cushing’s syndrome, thyroid disorders, and polycystic ovary syndrome. We found no concerning multicollinearity among these predictors based on Variance Inflation Factor (VIF) analysis with cut-off value of VIF <5 (Supplementary Table S1).”
Comments 6: Death could be a competing risk.
Response 6: Thank you for this insightful comment. We acknowledge that death can serve as a competing risk when studying incident ORCs. In response to your suggestion, we conducted additional analyses using competing risk models (Fine–Gray subdistribution hazard models) to account for death as a competing event. The results are presented in the supplementary materials (Supplementary Tables S5, S6, and S7) and summarized in the Results section as follows:
(page 9, lines 5-9) “Additionally, results from the competing risk models (Supplementary Table S5) were consistent with those from the conditional Cox models (Figure 2), supporting the robustness of our findings. The direction and magnitude of associations remained similar, indicating that accounting for the competing risk of death did not materially alter the observed relationships between AOM use and ORCs risk.”
(page 9, lines 14-22) “In the sensitivity analysis using competing risk models (Supplementary Table S6), results closely mirrored those from the conditional Cox models (Figure 3). The observed associations between AOM use and ORCs risk were stable across both approaches, suggesting that the risk of death had minimal impact on the observed relationships.
(page 10, lines 299-301) “). Similar findings were observed in the sensitivity analysis with competing risk models (Supplementary Table S7) except for a significant association between liraglutide use and reduced risk of ORCs (sHR = 0.71; 95% CI: 0.54–0.93; p = 0.014.).”
Comments 7: For cancer-specific analyses, if a patient developed BrCa first then CRC, would the patient only be included in the BrCa related analyses or in both BrCa and CRC? If former, other ORC except of the cancer of interest should also be considered as competing risks.
Response 7: Thank you for this thoughtful question. As described in the Methods section, we restricted the cohort to individuals who were free of any ORCs for at least six months following the index date (first AOM exposure date). Patients were then followed until the occurrence of their first incident ORCs, which was treated as the primary outcome. For example, if a patient developed BrCa first and subsequently CRC, only the BrCa event was considered, and the patient contributed to the BrCa analysis only. This approach aligns with our study design and objective. Because the analysis was limited to first events and follow-up was censored at the time of that first event, subsequent ORCs were not considered competing risks. As a result, a competing risks analysis was not deemed necessary from a statistical perspective. We have revised the ORCs Outcomes subsection and the Methods section for better clarification as follows:
(page 4 line 4 to page 5 line 9) “We examined the first incident ORCs, advanced stage ORCs, high grade ORCs, BrCa, advanced stage BrCa, high grade BrCa, CRC, advanced stage CRC, high grade CRC, ECa, advanced stage ECa, high grade ECa, and OCa. These cancers were selected based on consistent associations with excess body fat identified by the International Agency for Re-search on Cancer (IARC) [25]. Incident ORCs was identified using International Classification of Diseases for Oncology-3 (ICD-O-3) site and histology codes from the SEER data-base. BrCa was defined by site codes C50.0–C50.9 and histology codes 8500–8543. CRC included site codes C18.0–C20.9 and histology codes 8140–8147, 8210–8211, 8220–8221, 8260–8263, 8480–8481, and 8490. ECa was defined by site codes C54.0–C55.9 and histology codes 8140, 8262, 8323, 8380–8384, 8480–8482, and 8560. OCa was identified using site code C56.9 and histology codes 8440–8469. Cancer staging was based on the Ameri-can Joint Committee on Cancer (AJCC) system available in SEER. We assigned the first day of the cancer diagnosis month to enhance censoring accuracy, as SEER reports only the month and year of cancer diagnosis. Time-to-event was calculated in months starting from the index date (first AOM exposure). For patients who developed an ORCs, time-to-event was measured from the index date to the first ORCs diagnosis. Patients who did not develop ORCs were right-censored at the study end date (December 31, 2015) or at the time of loss to follow-up, whichever occurred first. Patients who remained ORCs-free and died before the study end date were right-censored at the time of death.”
Comments 8: One major concern is the underlying differences between exposed and non-exposed patients. Exposed patients were overweight or obese, yet unexposed patients might be normal weight or even under-weight. Although authors have applied propensity score matching to balance the potential confounding between the two groups, it might be helpful if the authors could conduct some sensitivity analyses among more homogeneous population, for example patients with diabetes, and add more discussions on it.
Response 8: Thank you for your valuable comment. As you mentioned, we initially addressed the differences between exposed and unexposed patients using 1:2 propensity score greedy nearest neighbor matching without replacement procedure to create comparable AOM groups with respect to observable characteristics. Following your suggestion, we conducted stratified analyses by diabetic status to address potential differences between exposed and non-exposed patients. Diabetes is an important factor that may influence both obesity and cancer risk. The results from these stratified analyses are presented in the supplementary materials (Supplementary Tables S4) and summarized in the Results section as follows:
(page 8 line 10 to page 9 line 5) ““In the sensitivity analysis (Supplementary Table S4) for the primary analysis, the association between AOM use and risk of ORCs was evaluated separately among women with and without diabetes. Among women with diabetes, AOM use was associated with a lower risk of several ORCs. Specifically, ever-use of AOM was associated with a 19% reduced risk of ORCs (aHR = 0.81; 95% CI: 0.73–0.89; p < .001). AOM use was also associated with a 34% reduction in advanced stage ORCs (aHR = 0.66; 95% CI: 0.53–0.83; p = <.001), a 23% lower risk of BrCa (aHR = 0.88; 95% CI: 0.78–0.99; p = 0.045), and a 23% reduced risk of CRC (aHR = 0.77; 95% CI: 0.62–0.95; p = 0.015). Additionally, AOM use was associated with a 34% reduction in advanced stage CRC (aHR = 0.66; 95% CI: 0.46–0.94; p = 0.023) and a 51% lower risk of OCa (aHR = 0.49; 95% CI: 0.28–0.86; p = 0.012). Among women without diabetes, AOM use was only associated with a 22% lower risk of ORCs (aHR = 0.78; 95% CI: 0.71–0.85; p < .001) and a 19% reduced risk of BrCa (aHR = 0.81; 95% CI: 0.73–0.92; p = <.001).”
Also, we have added further discussion to interpret these findings in the context of a more homogeneous population in the Discussion section as follows:
(page 12, lines 33-39) “The association between AOM use and ORCs risk was directionally consistent across subgroups, with stronger and statistically significant effects observed in women with diabetes. In this group, AOM use was linked to reduced risks of overall ORCs, advanced stage ORCs, BrCa, CRC, advanced stage CRC, and OCa This pattern mirrors the primary analysis closely. Among women without diabetes, associations were limited to ORCs BrCa. These findings suggest that the impact of AOMs on ORCs risk may be more evident among those with higher baseline metabolic risk.”
Comments 9: Minor comment: The title for 2.7 is wrong.
Response 9: Thank you for pointing this out. We have corrected the title for subsection 2.7, page 5 line 32, in the Methods section to accurately reflect its content. The revised title is now “Statistical Analysis”.
Again, we thank you very much for the thoughtful and constructive comments, which have significantly improved the quality of our manuscript. We believe that the revisions we have made address your concerns and strengthen the overall clarity and rigor of the study.
Reviewer 3 Report
Comments and Suggestions for Authors
The study examines the effect of anti-obesity drugs (AOM), especially phentermine and liraglutide, on obesity-related cancer (ORC) risk in women aged 65 years and older with SEER-Medicare data.
In general, an essential study for public health and factors affecting obesity-related cancer development,
Acceptable with a few minor corrections.
P values in the text can be given as p<0.001 for those with p < 0.0001.
P should be written lowercase p in the text.
Cox regression analyses were applied strongly in the study, and HRs were significant.
However, Kaplan-Meier curves (e.g. AOM users vs. non-users for BrCa) can be added. Differences in incidence over time can be assessed visually. The findings can be made more holistic if supported by the log-rank test. Kaplan-Meier Curves can be given to show how many patients had cancer at the end of 10 years and the projection of cancer at the end of 20 years, and the article can be made stronger.
There are too many repeated words in the text; if they are reduced, the text becomes simpler.
Author Response
Comments 1: P values in the text can be given as p<0.001 for those with p < 0.0001.
Response 1: Thank you for the suggestion. We have revised the manuscript accordingly and now report all p-values < 0.0001 as p < 0.001 throughout the text.
Comments 2: P should be written lowercase p in the text.
Response 2: Thank you for the feedback. We have updated the manuscript to use lowercase p throughout, as recommended.
Comments 3: Cox regression analyses were applied strongly in the study, and HRs were significant.
Response 3: Thank you for your positive feedback. We appreciate your recognition of the strength and significance of the Cox regression analyses in our study.
Comments 4: Kaplan-Meier curves (e.g. AOM users vs. non-users for BrCa) can be added. Differences in incidence over time can be assessed visually. The findings can be made more holistic if supported by the log-rank test. Kaplan-Meier Curves can be given to show how many patients had cancer at the end of 10 years and the projection of cancer at the end of 20 years, and the article can be made stronger.
Response 4: Thank you for the thoughtful suggestion. We did survival probability analyses and generated Kaplan–Meier curves for the primary analysis and included them in the supplementary materials (Supplementary Figure S1). We have also updated the Results section as follows:
(page 7, lines 7-13): “Supplementary Figure S1 presents Kaplan–Meier estimates of time to incident ORCs, stratified by AOM use in a 1:2 propensity score–matched cohort. Median follow-up duration varied across cohorts: 49.2 months for ORCs, 55.2 months for BrCa, 62.3 months for CRC, 63.3 months for ECa, and 64.3 months for OCa. Overall survival rates for incident ORCs differed significantly between AOM ever-users and never-users (log-rank test, p < 0.001), with ever-users showing a survival advantage. Similar trends were observed in the incident BrCa (p < 0.001) and CRC (p = 0.010), but not in ECa (p = 0.086) or OCa (p = 0.065).
Comments 5: There are too many repeated words in the text; if they are reduced, the text becomes simpler.
Response 5: Thank you for your valuable feedback. We conducted extensive editing throughout the manuscript sections, addressing repetitive wording and revising to improve clarity and conciseness.
Again, we thank you very much for the thoughtful and constructive comments, which have significantly improved the quality of our manuscript. We believe that the revisions we have made address your concerns and strengthen the overall clarity and rigor of the study.